# Prevalence, Antimicrobial Susceptibility, and Resistance Genes of Extended-Spectrum β-Lactamase-Producing *Escherichia coli* from Broilers Sold in Open Markets of Dakar, Senegal

**DOI:** 10.3390/microorganisms12112357

**Published:** 2024-11-19

**Authors:** Abdoulaye Cissé, Bissoume Sambe Ba, Ousmane Sow, Abdoul Aziz Wane, Issa Ndiaye, Cheikh Fall, Makhtar Camara, Yakhya Dieye

**Affiliations:** 1Pôle de Microbiologie, Institut Pasteur de Dakar, Dakar BP 220, Senegal; ablayecisse22@live.fr (A.C.); bsambe09@gmail.com (B.S.B.); ousmane.sow@pasteur.sn (O.S.); abdoul.wane@pasteur.sn (A.A.W.); issa.ndiaye@pasteur.sn (I.N.); cheikh.fall@pasteur.sn (C.F.); 2Laboratoire de Bactériologie-Virologie, Hôpital Aristide le Dantec, Dakar BP 3001, Senegal; camaramakhtar@yahoo.fr; 3Groupe de Recherche Biotechnologies Appliquées & Bioprocédés Environnementaux (GRBA-BE), École Supérieure Polytechnique, Université Cheikh Anta Diop, Dakar BP 5085, Senegal

**Keywords:** antimicrobial resistance, extended-spectrum β-lactamase (ESBL), *E. coli*, broilers, *bla_CTX-M_* genes, Senegal

## Abstract

Extended-spectrum β-lactamase-producing *Escherichia coli* (ESBL-*Ec*) poses a significant public health concern due to its widespread prevalence and resistance to multiple antibiotics. This study aimed to assess the prevalence, resistance profile, and carriage of ESBL-encoding genes in ESBL-*Ec* isolates from broilers in two markets of Dakar, Senegal. Sampling over a one-year period revealed that 61.7% of broilers carried ESBL-*Ec* in their cecum. Antibiotic susceptibility testing of 186 ESBL-*Ec* isolates showed high resistance to β-lactam antibiotics, including ampicillin, ticarcillin, and third-generation cephalosporins. Notably, all isolates were susceptible to imipenem. Multidrug resistance was frequent, with 91.4% of the isolates displaying this phenotype. Comparison between the two markets revealed variations in resistance to quinolones. PCR analysis detected *bla_CTX-M_* genes in all isolates, with *bla_CTX-M-1_* being the most prevalent subgroup. Additionally, *bla_TEM_* and *bla_OXA_* genes were found in 26.3% and 2.7% of isolates, respectively, while no *bla_SHV_* genes were detected. Combinations of ESBL genes were common, with *bla_CTX-M15_-bla_TEM_* being the most frequent. These findings highlight the widespread prevalence of ESBL-*Ec* in broilers from Dakar markets, indicating a potential risk of transmission to humans through the food chain. Effective surveillance and intervention strategies are crucial to mitigate the spread of antimicrobial resistance in humans and animals.

## 1. Introduction

The emergence of antimicrobial resistance (AMR) in bacterial pathogens is a growing global concern, as it can lead to increased morbidity and mortality rates, higher healthcare costs, and a reduction in the effectiveness of existing antimicrobial treatments [1]. The global burden of bacterial AMR was estimated at 5.7 million deaths and 156.1 million disability-adjusted life years attributable to antibiotic-resistant bacterial infections for 2019 [2]. This burden is higher in low- and middle-income countries (LMICs), particularly in sub-Saharan Africa [2]. Several strategies have been designed to address AMR globally [3]. Two of them that are particularly relevant to LMICs are capacity building for studying AMR and integrated surveillance of different sectors. Indeed, there is a lack of properly equipped laboratories and well-trained microbiologists in many LMICs. Consequently, quality data are highly needed in the human, animal, food, and environmental sectors in order to efficiently address the threat of AMR in LMICs. Surveillance is a key component of the global effort to tackle AMR. It relies on the isolation and detailed characterization of resistant bacteria. Importantly, AMR surveillance is concomitantly conducted in different sectors in order to possibly identify risk factors for acquiring and disseminating resistant bacteria. This integrated one-health approach has demonstrated its value in identifying the source of outbreaks, dissemination routes of clonal infections, and hotspots of selection of resistant bacteria [4].

β-lactam antibiotics are a cornerstone in both human and veterinary medicine, renowned for their broad-spectrum efficacy against a wide array of bacterial infections [5]. This class of antibiotics, characterized by the presence of a beta-lactam ring in their molecular structure, includes penicillins, cephalosporins, carbapenems, and monobactams. Their mode of action involves inhibiting the synthesis of bacterial cell walls, leading to cell lysis and death, making them highly effective in treating infections caused by Gram-positive and Gram-negative bacteria. Their effectiveness in human health, combined with generally favorable safety profiles, has made them a first-line treatment choice in many clinical settings [6]. The introduction of penicillin in the 1940s revolutionized medical practice, significantly reducing the morbidity and mortality associated with bacterial infections and ushering in the antibiotic era [7]. In veterinary medicine, beta-lactam antibiotics play a crucial role in maintaining animal health and ensuring the wellbeing of livestock, pets, and wildlife. They are used to treat infections, prevent disease outbreaks, and in some cases, promote growth in farm animals [8]. Ensuring the health of animals through the judicious use of antibiotics helps secure food safety and public health by preventing the transmission of zoonotic pathogens to humans [9]. However, the extensive use and, in some instances, misuse of beta-lactam antibiotics have led to the emergence of antibiotic-resistant bacteria [10]. This growing resistance poses a significant threat to both human and veterinary medicine, as it compromises the ability to treat common infections and conduct routine medical and surgical procedures safely. Resistance to β-lactams is primarily mediated by β-lactamases (BLs), which constitute a large and diverse family of enzymes that have in common the ability to inactivate β-lactams [5]. BLs are classified according to two schemes: the Ambler classification, that groups BLs into four classes (A–D) depending on the sequence of their active site [11], and the Bush-J classification, that divides BLs into three main groups based on their substrate and inhibitor profiles [12]. Among BLs, extended-spectrum β-lactamases (ESBLs) are the most important to public health. ESBLs were initially defined as BLs effective against third- and fourth-generation cephalosporins and monobactams, and are inhibited by β-lactamase inhibitors such as clavulanic acid, sulbactam, and tazobactam [13]. However, the emergence of ESBL variants resistant to the ancient inhibitors challenges this definition and, given their growing prevalence [14], stresses the need for novel inhibitors. The most common ESBL enzymes can be classified into four families: Bla_TEM_, Bla_SHV_, Bla_OXA_, and Bla_CTX-M_. The first known ESBL was derived from amino acid substitutions of Bla_TEM_ and Bla_SHV_ BLs that confer to these enzymes the ability to hydrolyze extended-spectrum cephalosporins [15]. Later on, the Bla_CTX-M_ enzymes emerged that were found to be largely disseminated worldwide and constitute by far the most prevalent ESBL family. The *bla_CTX-M_* gene family comprises five groups: CTX-M1, CTX-M-2, CTX-M-8, CTX-M-9, and CTX-M-25; each group includes several variants, the most prevalent being the CTX-M-15 variant, a member of the CTX-M1 group [16]. Bla_TEM_, Bla_SHV_, and Bla_CTX-M_ enzymes belong to class A of the Ambler classification, contrary to Bla_OXA_, the last of the four major ESBL families that are class D enzymes.

The importance of ESBL-producing bacteria in human and veterinary medicine makes these microorganisms ideal targets for AMR surveillance. The WHO developed a one-health surveillance program based on ESBL-producing *Escherichia coli* (ESBL-*Ec*). The program, called Tricycle, aims to provide a global picture by determining the prevalence of ESBL-*Ec* in the human, animal, and environmental sectors [17]. It uses standardized protocols that can be applied in laboratories with relatively limited equipment, allowing comparisons across countries, including developed nations and LMICs. In this study, we report the results of a one-year surveillance of ESBL-*Ec* in broilers sold in two open wet markets of Dakar, the capital of Senegal, as part of a pilot investigation using the Tricycle protocol. We describe the prevalence of ESBL-*Ec* carriage and, additionally, the molecular analysis of ESBL-encoding genes harbored by the isolated ESBL clones.

## 2. Materials and Methods

### 2.1. Sampling, Bacterial Isolation, and Identification

Broilers were purchased at two open wet markets, which are marketplaces where fresh food products, including meat, eggs, vegetables, and fruits, are sold in open, non-refrigerated spaces. The samples were collected according to the World Health Organization’s Tricycle protocol [18]. The two markets (Sandaga and Tilene), which are located within 5 km of the city center of Dakar, capital of Senegal, daily host broilers originating from different farms of the Dakar region. Sampling was conducted monthly over one year from September 2018 to August 2019. Each month, 10 birds were purchased from each market, totaling 240 pieces over the course of the study. The broilers were supplied by vendors who possess fixed canteens in the markets, with only one broiler obtained from a given vendor at any sampling round. The birds were slaughtered in the markets, and the intact ceca were collected in individual sterile plastic bags and placed in a cooler before shipment to the laboratory within two hours for immediate processing. For isolation of ESBL-*Ec*, a loopful (~10 mg) of each cecal content was directly streaked onto a MacConkey agar (Bio-Rad, Marnes-la-Coquette, France) plate supplemented with 4 µg/mL cefotaxime (Sigma, Burlington, MA, USA) that was incubated overnight at 37 °C. Five presumptive *E. coli* colonies (red/purple colonies) were randomly selected, re-isolated on the same selective plates, identified using standard microbiological tests and an API 20E system (bioMérieux, Craponne, France), and confirmed using a matrix-assisted laser desorption ionization time-of-flight (MALDI-TOF, Bruker, Bremen, Germany) mass spectrometry platform (Bruker Daltonik, Bremen, Germany).

### 2.2. Antimicrobial Susceptibility Testing and ESBL Confirmation

Antimicrobial susceptibility testing was performed using the Kirby Bauer disc diffusion method according to the European Committee on Antimicrobial Susceptibility Testing (EUCAST/CA-SFM) guidelines v1.1 2020 (https://www.sfm-microbiologie.org/wp-content/uploads/2020/04/CASFM2020_Avril2020_V1.1.pdf, accessed on 12 November 2024). The antibiotic disks (Bio-Rad, France) used included ampicillin (10 µg), amoxicillin + clavulanic acid (20 µg–10 µg), ticarcillin (75 µg), cefalotin (30 µg), cefotaxime (5 µg), ceftazidime (10 µg), cefepime (30 µg), aztreonam (30 µg), cefoxitin (30 µg), tetracycline (30 µg), nalidixic acid (30 µg), ciprofloxacin (5 µg), imipenem (10 µg), gentamicin (10 µg), and sulfamethoxazole + trimethoprim (1.25 µg–23.75 µg). The results were interpreted as susceptible, intermediate, or resistant. The double-disk synergy test was used to confirm the ESBL phenotype of the tested clones according to the EUCAST guideline on detection of resistance mechanisms version 2.0 (the last version so far). For this purpose, the cefotaxime, ceftazidime, and cefepime discs were placed at 1.5 cm from the one containing amoxicillin + clavulanic acid (EUCAST/CA-SFM 2020 guidelines). The ESBL phenotype was confirmed when the inhibition zone around a cephalosporin disc was increased toward the amoxicillin + clavulanic acid disc compared with the other sides. *E. coli* ATCC 25922 and *Klebsiella pneumoniae* ATCC 700603 were used as ESBL-negative and ESBL-positive reference strains, respectively.

### 2.3. Detection of ESBL Resistance Genes

Total bacterial DNA was extracted using a genomic DNA purification kit (QIAampDNA Mini Kit, QIAGEN, Courtaboeuf, France) according to the manufacturer’s recommendations. Detection of ESBL *bla_TEM_*, *bla_SHV_*, *bla_OXA1_*, and *bla_CTX-M_* genes was performed by PCR using the primers shown in Table 1. Samples positive for *bla_CTX-M_* were further analyzed to identify the subfamilies, including *bla_CTX-M-1_*, *bla_CTX-M-2_*, *bla_CTX-M-8_*, *bla_CTX-M-9_*, and *bla_CTX-M-25_* groups. Additionally, the presence of *bla_CTX-M-15_* variants was screened in all samples positive for *bla_CTX-M-1_*. The PCR assays were performed on an Eppendorf thermal cycler (Mastercycler X50a, Eppendorf, Montesson, France) using a program that included an initial denaturation at 95 °C for 3 min, followed by 35 cycles consisting of a denaturation at 94 °C for 1 min, a 1 min annealing at a temperature depending on the primers, and an elongation at 72 °C for 1 min. A final elongation at 72 °C for 7 min was included. The amplicons were analyzed by electrophoresis on a 1.5% agarose gel.

### 2.4. Statistical Analysis

Statistical analyses were carried out using IBM SPSS v28 software. Pearson’s chi-square was used to compare the antibiotic resistance between the ESBL-*Ec* isolates from the two markets, a *p* value < 0.05 being considered as statistically significant.

## 3. Results

### 3.1. Prevalence of Extended-Spectrum β-Lactamase-Producing Escherichia coli in Broilers

To survey for ESBL-*Ec* carriage in broilers, we conducted a prospective sampling in two wet markets in Dakar, the capital city of Senegal. The samplings were conducted monthly during a 1-year period between September 2018 and August 2019. Each market was visited monthly, with 20 broilers purchased at each visit. Of the 240 broilers (120 from each market) analyzed, 148 (61.7%, 74 from each market) carried ESBL-*Ec* in their cecum (Table 2). There was not a significant difference of the prevalence of ESBL-*Ec*-positive broilers according to market or sampling period.

### 3.2. Antibiotic Resistance Profile of Extended-Spectrum β-Lactamase-Producing Escherichia coli from Broilers

A total of 186 ESBL-*Ec* isolates were recovered, 98 (52.7%) and 88 (47.3%) from the Sandaga and Tilene markets, respectively (Table 3). All isolates were tested for susceptibility to 15 antibiotics belonging to 8 classes (Appendix A). As expected, resistance to β-lactam antibiotics was frequent, with all the isolates resistant to ampicillin and ticarcillin, which belong to the penicillin subfamily, and to the first- and third-generation cephalosporins cefalotin and cefotaxime, respectively (Table 3). Similarly, 83.9%, 84.4%, and 98.4% of the isolates were fully or intermediately resistant to ceftazidime (third-generation cephalosporin), cefepime (fourth-generation cephalosporin), and aztreonam (monobactam), respectively. In contrast, only 19.4% of the isolates were resistant or displayed an intermediate resistance to the second-generation cephalosporin cefoxitin (Table 3). Interestingly, all isolates were sensitive to imipenem (a carbapenem) (Table 3). Apart from β-lactams, resistance to the other families of antibiotics tested was frequent (above 65% of resistant or intermediate isolates) with the exception of gentamicin, for which 31.7% of the isolates were fully or intermediately resistant (Table 3). Not surprisingly, 91.4% (170/186) of the isolates were MDR, defined as clones fully or intermediately resistant to at least one molecule of at least three antibiotic classes (Appendix A). Lastly, when comparing the frequencies of resistance between the two markets, we found that isolates from Tilene market were significantly more frequently resistant to quinolones including nalidixic acid (80/88 [90.1%] and 71/98 [72.4%] of resistant or intermediately resistant isolates from Tilene and Sandaga, respectively; Pearson χ2 = 11.546, *p* = 0.003) and ciprofloxacin (66/88 [75.0%] and 57/98 [58.2%] of resistant or intermediately resistant isolates from Tilene and Sandaga, respectively; Pearson χ2 = 11.280, *p* = 0.004). In contrast, strains from Sandaga were significantly more resistant to ceftazidime (86/98 [87.8%] and 70/88 [79.5%] of resistant or intermediately resistant isolates from Sandaga and Tilene, respectively; Pearson χ2 = 6.333, *p* = 0.042). For all the other antibiotics tested, there was not a statistically significant difference between the isolates from the two markets.

### 3.3. Carriage of Extended-Spectrum β-Lactamase-Encoding Genes

We used PCR to detect ESBL-encoding genes, including *bla_CTX-M_*, *bla_TEM_*, *bla_OXA_*, and *bla_SHV_* families in the isolates (Figure 1, Table 4, and Appendix A). As expected, and consistently with their wide dissemination, *bla_CTX-M_* genes were found in all the isolates (Table 4). In contrast, *bla_TEM_* and *bla_OXA_* genes were detected in 26.3% (*n* = 49) and in 2.7% (*n* = 5) of the clones, respectively, while no *bla_SHV_* gene was detected in any of the isolates (Table 4). Regarding the subgroups of the *bla_CTX-M_* family, *bla_CTX-M-1_* genes were the most frequent, being present in 51.1% (*n* = 95) of the isolates (Table 4). Of these isolates, 61.1% (*n* = 58) harbored genes of the *bla_CTX-M-15_* subgroup. As for the other subgroups of the *bla_CTX-M_* family, *bla_CTX-M-9_*, *bla_CTX-M-2_*, and *bla_CTX-M-8_* were detected in 30.6% (*n* = 57), 28.5% (*n* = 53), and 10.2% (*n* = 19) of the isolates, respectively (Table 4), while no isolate contained a gene of the *bla_CTX-M-25_* subfamily. It should be noted that four isolates in which the CTX-M PCR was positive failed to generate an amplicon when analyzed for detection of the five subfamilies, suggesting the possible presence of a novel *bla_CTX-M_* variant in these clones (Table 4, Appendix A). Carriage of combinations of ESBL genes was frequent, with 41.9% (*n* = 78) of the isolates harboring two or three of these genes (Table 5). The most frequent combination was *bla_CTX-M15_*–*bla_TEM_*, found in 10.2% (*n* = 19) of the isolates (Table 5). Additionally, 9.8% (*n* = 18) of the isolates carried three ESBL genes, mostly corresponding to combinations of genes of the *bla_CTX-M_* and *bla_TEM_* families (Table 5). Carriage of two ESBL genes essentially consisted of combinations of CTX-M and TEM or of two genes of the *bla_CTX-M_* family (Table 5).

## 4. Discussion

In this study, we analyzed the prevalence, antibiotic resistance profile, and presence of ESBL-encoding genes among ESBL-*Ec* clones isolated from broilers sold in two open wet markets of Dakar, the capital city of Senegal. We found a high prevalence of ESBL-*Ec*-positive chickens, with 61.7% of the birds analyzed carrying this bacterium in their cecum. Additionally, and as expected, 83.9–100% of the isolates were resistant to the β-lactam antibiotics tested, including penicillins; monobactam; and first-, third-, and fourth-generation cephalosporins, the only exception being cefoxitin, a second-generation cephalosporin, for which only 19.4% of the tested clones displayed a reduced or an absence of susceptibility. Importantly, 91.4% of the isolates were MDR, displaying resistance to several antibiotic families. Regarding ESBL-encoding genes, the CTX-M group was the most frequent, including CTX-M-1, CTX-M-2, CTX-M-8, and CTX-M-9. In addition, 41.9% of the isolates carried at least two different ESBL genes as detected by PCR screening. It should be noted that, consistently with the Tricycle protocol [17], we directly inoculated cecal samples onto MacConkey agar plates containing 4 mg/L of cefotaxime. This concentration corresponds to the clinical breakpoint for this antibiotic. In a recent investigation, the European Union Reference Laboratory for AMR evaluated several protocols and found that the most sensitive and specific methodologies for detecting *ESBL-Ec* and AmpC-producing *E. coli* in cecal samples consisted of an initial pre-enrichment in buffered peptone water followed by inoculation onto MacConkey agar supplemented with 1 mg/L cefotaxime [21]. If used in our samples, this procedure would likely reveal a higher prevalence of ESBL-*Ec*.

The high prevalence of ESBL-*Ec* we found in the broilers was not surprising since these bacteria are widespread globally, not only in animals, but also in human, food, and environmental reservoirs. *E. coli* is a ubiquitous bacterial species, primarily serving as a commensal organism in the intestinal tracts of humans and animals. Due to its widespread presence, it serves as a reliable indicator of fecal contamination in the environment. *E. coli* contributes positively to host health by aiding in digestion, producing vitamins, and preventing colonization by pathogenic bacteria [22,23,24]. However, certain pathogenic strains of *E. coli* can cause a range of infections, including urinary tract, gastrointestinal, pulmonary, and bloodstream infections. These characteristics make *E. coli* a significant vector for the dissemination of AMR. Recognizing this, the WHO established an AMR surveillance program known as “Tricycle”. This program monitors the prevalence of ESBL-*Ec* across four domains: (i) bloodstream infections in humans, (ii) fecal carriage in pregnant women, (iii) cecal carriage in broiler chickens, and (iv) wastewater and surface waters impacted by human and/or animal activity [17]. A key feature of the Tricycle program is its use of standardized, straightforward protocols that can be conducted in microbiology laboratories with limited equipment, enabling cross-comparisons across high-income regions and LMICs worldwide. As of 2023, the Tricycle program was being implemented in over 19 countries. Preliminary reports indicate a high prevalence of ESBL-*Ec* across all sectors studied. In broilers, the reported prevalence of ESBL-*Ec* was 67.1% in Indonesia [25], 57% in Madagascar [26], and 38.6% in Nepal [27], figures comparable to those found in our study in Senegal. In humans, the prevalence of ESBL-*Ec* fecal carriage varied significantly: 40% in Indonesia [25], 30% in Madagascar [26], 22.3% in Benin [28], and 15% in Nepal [27]. These samples, collected from healthy pregnant women as per the Tricycle protocol, are intended to reflect community prevalence. For comparison, a recent study in the Central African Republic found a high fecal carriage rate of ESBL-*Enterobacterales* (85.3%) in children aged 0–5 years admitted to the Pediatric University Hospital Complex in Bangui, with the majority carrying ESBL-*Ec* and, to a lesser extent, ESBL-producing *K. pneumoniae* [29]. While the protocol in this study differed slightly from Tricycle in terms of ESBL-*Ec* selection, these results illustrate the variability of ESBL-*Ec* carriage prevalence in human populations. This variability is further supported by a retrospective meta-analysis of 66 studies up to July 2015, covering 28,909 individuals [30]. The analysis reported an overall 14% prevalence of ESBL-*Enterobacterales* carriage in healthy individuals across the six WHO regions, with an annual increase of 5.38%. The highest prevalence rates were found in the Western Pacific (46%), Southeast Asia (22%), and Africa (22%).

Unsurprisingly, wastewater samples exhibit the highest prevalence of ESBL-*Ec*. Reports from Benin [28], Ghana [31], Indonesia [25], Madagascar [26], and Nepal [27] indicate that 91–100% of wastewater samples from markets, slaughterhouses, or hospitals were contaminated with ESBL-*Ec*. In contrast, surface waters not directly impacted by human or animal activity show significantly lower prevalence and concentration of ESBL-*Ec*. These findings are consistent with a study we conducted on environmental samples from Ouagadougou, Burkina Faso, where 84.1% of wastewater samples from a municipal treatment plant and 62.1% of manure samples from livestock markets tested positive for ESBL-*Ec*, compared with only 39.7% of runoff water samples [32] (REF). Overall, these findings highlight human activity as a major driver of ESBL-*Ec* dissemination in the environment.

Several factors contribute to the spread of ESBL-producing bacteria, especially in LMICs, including poor hygiene, inadequate infection control practices in hospitals and clinics, limited surveillance and laboratory capacity, antibiotic misuse and overuse in both human and veterinary medicine, and, most importantly, the frequent use of antibiotics in livestock production that contributes to disseminating resistant bacteria via the food chain [33]. In this regard, chickens represent an important dissemination vehicle since they are the most common and most consumed meat worldwide. In Senegal, poultry farming is a significant activity, with production facilities corresponding to small-scale family units, semi-intensive farming systems, and a few large-scale commercial farms [34]. Family units are common and mainly for family consumption and limited-scale commercialization in the neighborhood. They typically use makeshift cages installed in courtyards or on terraces, which favors close contact between poultry and humans [34]. Contrary to family production, semi-intensive units and large commercial farms are destined to supply supermarkets or open markets in medium-sized to big cities. The use of antimicrobials as therapeutic or prophylactic agents as well as growth promoters is common in all the chicken production systems, although accurate data on its magnitude and the types of molecules used are lacking [35]. An important issue related to the use of antimicrobials in farming is the absence of regulation and of controls in the operation units. This favors the introduction of fraudulent molecules and is associated with the use of antimicrobials without veterinary prescription. All these factors contribute to the selection of resistant bacteria that can be disseminated through the food chain. A recent survey in 222 semi-intensive peri-urban chicken farms in Senegal reported that the main drivers of antimicrobial use are the unawareness about AMR and poor biosecurity practices [35].

## 5. Conclusions

Our study revealed a high prevalence of ESBL-*Ec* in broiler chickens sold at two open wet markets in Dakar, the capital of Senegal. This prevalence may be linked to the use of antimicrobials as growth promoters in poultry farming. Most of the isolated strains carried multiple ESBL genes, with variants of the CTX-M family being the most common. Additionally, these strains frequently exhibited resistance to other classes of antibiotics. Given the critical role of β-lactam antibiotics in both human and veterinary medicine, a robust surveillance system is essential to mitigate the potential risk of transmission of pathogenic ESBL-*Ec* through the consumption of broilers and chicken-derived food products. More broadly, in the global context of the AMR crisis, coordinated efforts are required to establish, implement, and enforce regulations governing antimicrobial use. This includes sustained, multisectoral surveillance, enhanced training for microbiologists, and increased public awareness in LMICs.

## Figures and Tables

**Figure 1 microorganisms-12-02357-f001:**
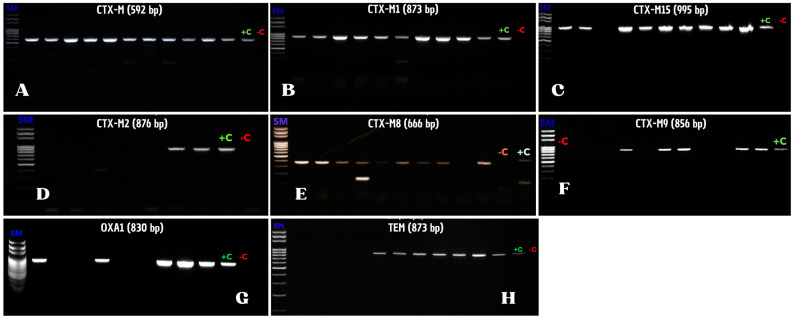
Sample results of PCR amplification of ESBL genes. Genomic DNAs of the ESBL isolates were used as templates for PCR reactions with primers that amplify *bla_CTX-M_* (**A**), *bla_OXA_* (**G**), *bla_TEM_* (**H**), and *bla_SHV_* gene families (see Table 1). Samples positive for the *bla_CTX-M_* family were subsequently analyzed for detection of genes belonging to the *bla_CTX-M1_* (**B**), *bla_CTX-M2_* (**D**), *bla_CTX-M8_* (**E**), *bla_CTX-M9_* (**F**), and *bla_CTX-M25_* subfamilies. Similarly, samples positive for *bla_CTX-M1_* were screened to identify *bla_CTX-M15_* variants (**C**). SM, DNA ladders; +C, positive control; −C, negative control.

**Table 1 microorganisms-12-02357-t001:** Characteristics of primers used in this study.

Target Genes	Primer Sequences	Amplicon Size (bp)	Annealing Temperature	References
*bla_TEM_*	F: TTGGGTGCACGAGTGGGTTA R: TAATTGTTGCCGGGAAGCTA	873	55 °C	[19]
*bla_SHV_*	F: TCGGGCCGCGTAGGCATGAT R: AGCAGGGCGACAATCCCGCG	628	52 °C	[19]
*bla_OXA1_*	F: ATGAAAAACACAATACATATC R: AATTTAGTGTGTTTAGAATGG	830	56 °C	[20]
*bla_CTX-M_*	F: ATGTGCAGYACCAGTAARGTKATGGC R: TGGGTRAARTARGTSACCAGAAYSAGCGG	592	55 °C	[19]
*bla_CTX-M-1_* group	F: GGTTAAAAAATCACTGCGTC R: TTACAAACCGTYGGTGACGA	873	50 °C	[19]
*bla_CTX-M-15_*	F: CACACGTGGAATTTAGGGACT R: GCCGTCTAAGGCGATAAACA	995	50 °C	[19]
*bla_CTX-M-2_* group	F: ATGATGACTCAGAGCATTCGCCGC R: TCAGAAACCGTGGGTTACGATTTT	876	56 °C	[19]
*bla_CTX-M-8_* group	F: TGATGAGACATCGCGTTAAG R: TAACCGTCGGTGACGATTTT	666	52 °C	[19]
*bla_CTX-M-9_* group	F: GTGACAAAGAGAGTGCAACGG R: ATGATTCTCGCCGCTGAAGCC	856	55 °C	[19]
*bla_CTX-M-25_* group	R: AACCCACGATGTGGGTAGC R: CCTCGCTGTGCTTGTATCC	327	52 °C	[19]

F, forward primer; R, reverse primer.

**Table 2 microorganisms-12-02357-t002:** Prevalence of extended-spectrum β-lactamase-producing *Escherichia coli* in broilers.

Markets	Number of Samples	ESBL-*Ec* Positive Samples (%)
Sandaga	120	74 (61.7)
Tilene	120	74 (61.7)
Total	240	148 (61.7)

ESBL-*Ec*, extended-spectrum β-lactamase-producing *Escherichia coli*.

**Table 3 microorganisms-12-02357-t003:** Resistance of extended-spectrum β-lactamase-producing *Escherichia coli* isolates to antibiotics tested in this study.

Antibiotic Families	Antibiotics	Resistance * N (%)
β-lactams	Ampicillin	186 (100)
	Ticarcillin	186 (100)
	Amoxicillin + clavulanic acid	0
	Cefalotin	186 (100)
	Cefotaxime	186 (100)
	Ceftazidime	156 (83.9)
	Cefepime	157 (84.4)
	Aztreonam	183 (98.4)
	Cefoxitin	36 (19.4)
	Imipenem	0
Quinolone	Nalidixic acid	151 (81.2)
Fluoroquinolone	Ciprofloxacin	123 (66.1)
Aminoglycoside	Gentamicin	59 (31.8)
Tetracycline	Tetracycline	166 (82.2)
Sulfonamide	Trimethoprim + sulfamethoxazole	139 (74.8)

* This includes isolates with a resistant or intermediate phenotype. N, number of isolates.

**Table 4 microorganisms-12-02357-t004:** Presence of extended-spectrum β-lactamase-encoding genes in ESBL-producing *Escherichia coli*.

ESBL Genes			Nb Isolates (%)
*bla_CTX-M_* family	*bla_CTX-M-1_* group	*bla_CTX-M15_* variant	58 (31.2%)
		Non-*bla_CTX-M15_* variant *	37 (19.9%)
	*bla_CTX-M2_* group		53 (28.5%)
	*bla_CTX-M8_* group		19 (10.2%)
	*bla_CTX-M9_* group		57 (30.6%)
	*bla_CTX-M25_* group		0
	Other *bla_CTX-M_* **		4 (2.2%)
*bla_TEM_* family			49 (26.3%)
*bla_OXA_* family			5 (2.7%)
*bla_SHV_* family			0

Nb isolates, number of PCR-positive isolates for the corresponding genes; *, Isolates with a CTX-M1-positive and a CTX-M15-negative PCR; **, Isolates with a CTX-M-positive PCR but negative for reactions detecting the five gene subfamilies of the CTX-M family.

**Table 5 microorganisms-12-02357-t005:** Single or combination of extended-spectrum β-lactamase-encoding genes carried by ESBL-producing *Escherichia coli*.

ESBL Gene Combination	Nb Isolates (%)
*bla_CTX-M1_ + bla_CTX-M8_ + bla_CTX-M9_*	4 (2.2%)
*bla_CTX-M15_ + bla_CTX-M8_ + bla_TEM_*	4 (2.2%)
*bla_CTX-M15_ + bla_OXA_ + bla_TEM_*	3 (1.6%)
*bla_CTX-M15_ + bla_CTX-M8_ + bla_TEM_*	2 (1.1%)
*bla_CTX-M15_ + bla_CTX-M2_ + bla_CTX-M9_*	1 (0.5%)
*bla_CTX-M15_ + bla_CTX-M2_ + bla_TEM_*	1 (0.5%)
*bla_CTX-M15_ + bla_CTX-M9_ + bla_TEM_*	1 (0.5%)
*bla_CTX-M1_ + bla_CTX-M2_ + bla_TEM_*	1 (0.5%)
*bla_CTX-M2_ + bla_CTX-M9_ + bla_TEM_*	1 (0.5%)
*bla_CTX-M15_ + bla_TEM_*	19 (10.2%)
*bla_CTX-M1_ + bla_TEM_*	9 (4.8%)
*bla_CTX-M2_ + bla_TEM_*	6 (3.2%)
*bla_CTX-M1_ + bla_CTX-M9_*	6 (3.2%)
*bla_CTX-M15_ + bla_CTX-M2_*	5 (2.7%)
*bla_CTX-M15_ + bla_CTX-M8_*	5 (2.7%)
*bla_CTX-M1_ + bla_CTX-M8_*	2 (1.1%)
*bla_CTX-M2_ + bla_CTX-M9_*	2 (1.1%)
*bla_CTX-M2_ + bla_OXA_*	2 (1.1%)
*bla_CTX-M15_ + bla_CTX-M9_*	1 (0.5%)
*bla_CTX-M8_ + bla_CTX-M9_*	1 (0.5%)
*bla_CTX-M8_ + bla_TEM_*	1 (0.5%)
*bla_CTX-M9_ + bla_TEM_*	1 (0.5%)

## Data Availability

The original contributions presented in this study are included in the article/Appendix A. Further inquiries can be directed to the corresponding author.

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
