# Peer review of "Prevalence, Antimicrobial Susceptibility, and Resistance Genes of Extended-Spectrum β-Lactamase-Producing Escherichia coli from Broilers Sold in Open Markets of Dakar, Senegal"

_microorganisms, 2024, doi:10.3390/microorganisms12112357_

Round 1

Reviewer 1 Report

Comments and Suggestions for Authors

The study investigated the prevalence, antibiotic resistance, and gene carriage of extended-spectrum β-lactamase-producing Escherichia coli (ESBL-Ec) in broilers from open markets in Dakar, Senegal. It does provide some valuable insights and emphasize the importance of enhanced surveillance. However, some issues in the article are not clearly described. Please revise and supplement the text according to the following specific suggestions.

1.       Lines 23-24, Please specifically explain the variations in resistance to quinolones between the two markets. Otherwise, kindly remove this sentence from the abstract.

2.       Lines 28-29, How do your results demonstrate “a potential risk of transmission to humans through the food chain”? Please explain or modify accordingly.

3.       Keywords: It is recommended to provide a maximum of six keywords; please remove the last two.

4.       Line 103, Are these “given vendors” fixed? Please explain.

5.       What is the reference strain for antimicrobial susceptibility testing? Is it E. coli ATCC 25922? Please add.

6.       Lines 145-146, 175-178, Why was resistance to quinolones compared only between the two markets? Are there no differences in resistance to other antibiotics, particularly β-lactams? Please explain.

7.       Table 2, Please revise the subtitle to emphasize the carriage rates.

8.       Table 3, Please change “Cyclin” and “Folate pathway inhibitor” to “tetracycline” and “Sulfonamide”.

9.       Table 4, This table is a bit unclear. What does "Nb" refer to? What do "* CTX-M1-positive, CTX-M15-negative isolates" and "** CTX-M-positive, CTX-M1-negative isolates" mean? Please remake the table or remove it.

10.   Please add a conclusion paragraph at the end of the Discussion.

11.   Please standardize the writing of Escherichia coli and pay attention to the italics.

Comments on the Quality of English Language

The English could be improved to more clearly express the research

Reviewer 2 Report

Comments and Suggestions for Authors

I would like to congratulate the authors on this well-written manuscript. The issue of Extended-Spectrum Beta-Lactamase (ESBL) is indeed a rising problem in healthcare. The authors have followed standard operating procedures (SOP) meticulously throughout their research. The sampling methods were clearly outlined, and the methodology was well articulated.

I have one suggestion for improvement. Since the sampling was conducted in 2018 and 2019, the authors have referred to 2020 guidelines. To make the findings more relevant to 2024, I recommend that they incorporate the most up-to-date EUCAST guidelines and breakpoints available. This adjustment will enhance the applicability of the manuscript and its relevance in current clinical contexts. 

Author Response

Authors reply in the attached file.

Reviewer 3 Report

Comments and Suggestions for Authors

Dear authors

Thanks for your work and presentation.

Many comments should be revised and considered before accepting the manuscript;

  1. In materials and methods section:
  • Line 92, what is the meaning of wet markets?!
  • Line 98, the term “poultry” should be replaced by “broilers”.
  • The exact number of the examined samples should be mentioned.
  • The source and concentration of antibiotic discs should be mentioned.
  • Line 125, the full name of K. pneumoniae should be mentioned at first.
  1. What about the photos of PCR results?!
  2. The discussion should be deeper and supported with references. For instance, the prevalence or the global picture of ESBL-Ec should be deeply discussed.
  3. The final conclusion of the manuscript should be mentioned at the end of the manuscript before the recommendation (270-273).

Best wishes

Author Response

Authors' reply in the attached file.

Round 2

Reviewer 1 Report

Comments and Suggestions for Authors

The authors have revised and responded to the comments as requested, and the manuscript is ready for publication.